# No Time to Age: Uncoupling Aging from Chronological Time

**DOI:** 10.3390/genes12050611

**Published:** 2021-04-21

**Authors:** Dana Larocca, Jieun Lee, Michael D. West, Ivan Labat, Hal Sternberg

**Affiliations:** 1DC Biotechnology, Alameda, CA 94502, USA; 2AgeX Therapeutics Inc., Alameda, CA 94501, USA; jlee@agexinc.com (J.L.); mwest@agexinc.com (M.D.W.); ilabat@agexinc.com (I.L.); hsternberg@agexinc.com (H.S.)

**Keywords:** aging, cellular clocks, reprogramming, development, epigenetics, DNA methylation, telomeres, transposable elements, longevity, regeneration

## Abstract

Multicellular life evolved from simple unicellular organisms that could replicate indefinitely, being essentially ageless. At this point, life split into two fundamentally different cell types: the immortal germline representing an unbroken lineage of cell division with no intrinsic endpoint and the mortal soma, which ages and dies. In this review, we describe the germline as clock-free and the soma as clock-bound and discuss aging with respect to three DNA-based cellular clocks (telomeric, DNA methylation, and transposable element). The ticking of these clocks corresponds to the stepwise progressive limitation of growth and regeneration of somatic cells that we term somatic restriction. Somatic restriction acts in opposition to strategies that ensure continued germline replication and regeneration. We thus consider the plasticity of aging as a process not fixed to the pace of chronological time but one that can speed up or slow down depending on the rate of intrinsic cellular clocks. We further describe how germline factor reprogramming might be used to slow the rate of aging and potentially reverse it by causing the clocks to tick backward. Therefore, reprogramming may eventually lead to therapeutic strategies to treat degenerative diseases by altering aging itself, the one condition common to us all.

## 1. Introduction

Whether or not aging is programmed or is the result of accumulated damage has been the subject of debate for many decades. Early twentieth-century theories favoring programming proposed that aging was a consequence of differentiation as put forth by Minot [1] or factors leading to the cessation of growth as proposed by Bidder [2]. The evolution of a genetic program for aging seemed unlikely because there is little selective pressure for gene variants once reproductive maturity is reached. However, Williams proposed that aging is the result of antagonistic pleiotropy, where mutations that are adaptive for fitness in early life but deleterious after reproduction would accumulate in the gene pool because there is little selective pressure to eliminate them [3]. In contrast, aging theories of the late twentieth century focused largely on “wear and tear”, suggesting aging results from the stochastic accumulation of damage, such as cellular waste, oxidative damage, DNA mutations, and misfolded proteins [4,5]. However, the vast range of adult lifespans among animal species from hours (mayfly), days (fruit fly), or weeks (dwarf pygmy goby) to as long as centuries (Aldabra giant tortoise, bowhead whale, Greenland shark), as well as examples of indefinite lifespans with no apparent intrinsic aging (hydra, planaria, sponge, mussel), suggests that underlying genetic programs must play a pivotal role [6,7]. Indeed, the wide variation in lifespan even between closely related species underscores the notion that aging is plastic and depends upon a balance between intrinsic cellular strategies that maintain genetic and epigenetic integrity and the effects of “wear and tear”. We have previously described the stepwise progressive loss of regenerative germline strategies in mortal somatic cells (somatic restriction) as an underlying cause of organismic decline associated with aging [8]. In contrast, we see in the regeneration of the immortal germline the potential for life to be uncoupled from time, and therefore free from aging. The fact that there can be an uncoupling of life from the fixed progression of chronological time as seen in the germline hints at the basis by which aging can be accelerated, decelerated, and perhaps most astonishingly reversed altogether.

In his germ plasma theory, the 19th-century biologist, August Weismann, described a division of labor in cellular life that enabled the evolution of complex multicellular life forms that make up the tremendous diversity of life we see today [9]. To make the leap from simple unicellular organisms to complex metazoans, life evolved into two fundamentally different types of cells consisting of the mortal cells of the soma or body, which eventually ages and dies, and the immortal self-renewing germline cells that carry the hereditary information forward to the next generation. Well over half a century before the discovery of cellular senescence, DNA, telomeres, and telomerase, Weismann predicted the limited cellular lifespan of somatic cells in contrast to cells of the germline, which have no intrinsic end to their replicative lifespan. He further asserted that “there is nothing inherent in life that implies death [9].” Aristotle wrote, “Things which are always are not, as such, in time nor is there being measured by time … none of them is affected by time, which indicates they are not in time.” In this sense, Weismann understood that life is not bound by time. To illustrate the evolutionary split of the soma and germline, Weismann described two genera of volvox, one of the simplest multicellular organisms (Figure 1). *Pandorina morum* consists of a simple ball of identical cells (Figure 1A) that reproduces by budding off smaller clusters of cells, and *Volvox minor* consists of a hollow sphere of cells (Figure 1B) that contains the simpler balls of germline cells within it. As the germline clusters grow larger, they trigger the sphere to wither and die and thus release the next generation of volvox. Accordingly, we see in the evolution of volvox, cells that exhibit a division of labor into immortal germline (embryonic balls of cells) and mortal soma (hollow sphere of cells). The germline can forever renew itself, giving birth to young organisms at each generation, and thus, we refer to germline cells as clock-free and somatic cells as clock-bound. Indeed, we can trace an unbroken chain of cell division from all life on earth going back 4 billion years to converge at the base of the evolutionary tree to a single cell, LUCA, the “last universal common ancestor” [10]. In contrast, somatic cells lose their robustness with time exhibiting the hallmarks of aging [11], which eventually lead to organismal degeneration and an exponential increase in the probability of death with the passage of time [12]. In this review, we will explore the nature of three somatic cellular clocks (telomeric, DNA methylation (DNAm), and transposable element (TE)) and the germline strategies used to become clock-free or unbound from chronological time. We also consider the relationship of aging to development and we describe remarkable examples of phenotypic plasticity seen in lower multicellular organisms that appear to reverse development. Finally, we discuss how this plasticity can be revived in higher animals using cell reprogramming technology aimed at conferring germline qualities onto the soma and how these methods might be applied to reverse aging and extend health span and lifespan.

## 2. Nature of Somatic Cellular Clocks: Telomeric Clock

The telomeric clock that limits the number of cell divisions of somatic cells was accurately predicted by Weismann. He noted that “death takes place because a worn-out tissue cannot forever renew itself and because a capacity for increase by means of cell division is not everlasting but finite” [9]. However, it would be a half-century before Leonard Hayflick performed definitive experiments that proved the limited lifespan of somatic cells [13]. Previously, it had been widely accepted that somatic cells were immortal. In 1912, Carrel had demonstrated the continuous passage of chick heart cells for over three months [14], and indeed, the cells were passaged for more than three decades, outliving Carrel himself [14]. Carrel’s influence in the press and Nobel laureate status was so strong that the lack of reproducibility in other laboratories was not enough to keep his theory of cellular immortality from becoming the prevailing view. However, in 1961, Hayflick reported conclusive evidence that human fibroblasts were limited to about 50 population doublings before reaching replicative senescence [13]. This limited replication capacity of somatic cells is referred to as the Hayflick limit. He elegantly proved that the cessation of replication was intrinsic to the cells and not due to external factors by coculture of late passage male cells with early passage female cells and demonstrating that the young female cells continued to grow well after the male cells reached senescence [15]. It is now well established that virtually all normal human cells have a finite replicative capacity [16]. There is much speculation as to how Carrel was able to maintain continuous replication of his cultures for decades, including inadvertent or deliberate spiking with new cells. One source of external cells may have been the chick embryo extract that was used early on to maintain the health of the cultures. Alternatively, given what we now know about cellular reprogramming using germline factors, it is interesting to speculate that secreted factors in Carrel’s embryo extracts may have immortalized his cell cultures.

Hayflick’s experiment confirmed Weismann’s prior conjecture about the limited replicative lifespan of somatic cells and demonstrated the phenomenon of cellular senescence, which left cells in a seemingly irreversible suspension of the cell cycle. The Hayflick clock can be suspended temporarily by inducing quiescence or by storage at low temperature because it is a measure of replication cycles [15,17]. The clock mechanism was independently proposed in the early seventies by Olovnikov and Watson, who each theorized that the ends of DNA would necessarily shorten each time it replicated due to the end replication problem [18,19]. Olovnikov further hypothesized that enzymatic repair could preserve or elongate the ends of telomeres to prevent the shortening of chromosomal telomere ends at each cell division [18,20]. This hypothesis was validated in the 1990′s with the cloning of human telomerase reverse transcriptase (*hTERT*), the gene encoding the catalytic component of the enzyme responsible for telomere maintenance, and by showing that forced expression of *hTERT* prevented senescence, thus immortalizing the cells [21]. Remarkably, for both the germline and pluripotent embryonic stem cells, the telomeric clock is suspended, but for somatic cells, the clock begins to tick not long after the initiation of embryonic development [22,23]. We have previously described the loss of immortal regeneration and replication as the pluripotency transition (PT) [8]. It is the first step in a series of life history transitions, termed “somatic restriction”, that lead to the loss of regenerative capacity characteristic of aging [8]. Another type of immortal cell, the cancer cell, also expresses telomerase, as shown using a telomerase activity assay [23]. Indeed, the evolutionary advantage of shutting off telomerase may be as a protection against cancer, resulting in greater reproductive fitness in early life. However, progressive shortening of telomeres eventually triggers cell senescence which in later life contributes to lack of regeneration, immune system failure, and chronic inflammation due to senescence-associated secretory products (SASP) [24]. Thus, the progressive restriction of germline strategies for immortal regeneration is consistent with the antagonistic pleiotropy theory of the evolution of aging.

Evidence for the connection between the telomeric clock and aging comes from the correlation of telomere shortening with age in humans and experimental animals, simulation of human-like aging in mice lacking telomerase, and from conditions such as progeria and HIV infection that result in rapid telomere shortening and accelerated aging [25,26]. Although there is some controversy arising from the variability of telomere measurement techniques and cell type chosen for analysis, many studies demonstrate a correlation of telomere attrition with aging in humans [27,28,29]. Moreover, data from a variety of vertebrate model animals demonstrate that telomere shortening is associated with human-like age-related disease and lifespan. For example, the short-lived African killifish has served as a useful model of telomere dysfunction and aging [30]. Mice normally have much longer telomeres than humans; however, successive generations of single Terc (mouse catalytic component of telomerase) knockout or double knockout of Wrn (the helicase gene that is mutated in Werner syndrome) and Terc show telomere attrition and age-related conditions that more closely model human aging [31,32]. Importantly, reactivation of telomerase in this model restores degenerative damage in multiple systems, including neurodegeneration [33]. Furthermore, an accelerated telomeric clock resulting from defects in telomere maintenance is associated with several human diseases that mimic rapid aging, including Hutchinson–Gilford progeria syndrome (HGPS), Werner syndrome, and dyskeratosis congenita (DKC) [25]. Additional association of the telomeric clock with age-associated disease comes from combined mouse models that demonstrate the contributing influence of telomere shortening on diabetes and cardiomyopathy in Duchenne muscular dystrophy [34,35]. In addition, immune deficiencies in the elderly and HIV patients have been linked to the Hayflick limit of T cells [36]. Indeed, relatively young HIV patients show a similar degree of T cell telomere attrition as centenarians [26]. Although long-lived post-mitotic cells in non-proliferative tissues such as the brain, skeletal muscle, fat, and heart were not initially thought to be affected by the telomere clock, telomere shortening has been identified more recently in these tissues, which may be triggered by SASP from other cells [37]. Telomere attrition has also been proposed to contribute to both the timing of parturition and death through the onset of inflammation [38]. The telomeric clock can also be slowed, for example, by experimental overexpression of telomerase which has been shown to extend lifespan even when given late in life [39]. Interestingly, while the telomere length at birth is highly variable and has not been found to correlate with lifespan, the rate of telomere attrition correlates well with lifespan in a wide variety of bird and mammal species [40]. Moreover, mice generated from ES cells with hyper-long telomeres show an increase in lifespan, are leaner and have a decrease in metabolic aging [41]. Importantly, environmental interventions such as calorie restriction and exercise have been shown to decelerate the telomeric clock in humans [42,43]. Taken together, these data support the notion that the loss of telomerase near the pluripotency to embryonic transition (PT) correlates with the onset of mortality of the soma and a clock-bound process that contributes to the eventual aging and death of the body. We will next discuss two additional clocks, the epigenetic or DNA methylation (DNAm) clock and the transposable element (TE) clock.

## 3. Nature of Somatic Cellular Clocks: Epigenetic (DNAm) Clock

The epigenetic clock represents changes in the methylation pattern of CpG sites in the genome that occur in somatic cells over time in a predictable manner. Methylome changes also occur in the germline with age but are reset upon fertilization [44]. Other epigenetic changes such as reduced global heterochromatin, increase in senescence-associated heterochromatin foci (SAHF), changes in histone marks, and relocation of chromatin-modifying factors are recently reviewed elsewhere [45]. The DNAm clock, which likely is influenced by and influences other epigenetic changes, is currently the most advanced in its ability to predict chronological age using data from a variety of tissues [46,47]. Initially, global DNA methylation was found to decrease with age [48]. However, microarray technology enabled the assessment of site-specific CpG methylation, which led to the first methylation clocks based on large data sets of blood samples [46]. Although the development of DNAm clocks has included clocks based on only one or a few loci [48,49,50], the more widely used multi-tissue clock developed by Horvath measures DNAm at 353 CpG sites has greater accuracy with a median error of chronological age prediction of fewer than 4 years [47]. Like the telomeric clock, the DNAm clock starts at the beginning of development, soon after the differentiation of pluripotent embryonic stem cells when cells transition from immortal germline to mortal somatic cells. Although the DNAm clock is linked to development, it is not necessarily in synchrony with differentiation [51]. The developing retina, for example, contains cells representing various stages of differentiation, but all have the same DNAm age [51]. Several placental and cord blood DNAm clocks have been developed that estimate gestational age at birth [52,53,54] to various degrees of accuracy depending on the study design [55]. They may be useful for the correlation of gestational age acceleration with health outcomes. Distinct DNAm patterns have also shown value in delineating the fetal from the adult state [56]. Importantly, hematopoietic stem cell transplantation patients have shown that the recipient’s blood cells continue to reflect the DNAm age of the donor despite large differences between donor and recipient ages, indicating that DNAm clock age is a cell-intrinsic property [57]. Remarkably, a lack of age acceleration in human hematopoietic cells that are transplanted into mice provides further evidence for a cell-intrinsic DNAm clock [58]. Many other DNAm clocks have been developed since the initial multi-tissue clock, including a clock for mouse tissues [59] as well as for cells in culture [60]. The availability of a non-biased measurement of biological aging in experimental models will undoubtedly increase our understanding of aging and our ability to screen for compounds that can slow or reverse the process.

Many questions remain about the nature of DNAm clocks, their underlying molecular mechanism, and their relationship as a cause of aging or effect of other biological processes that bind somatic cells to time. So far, we have seen that two clocks, telomeric and DNAm, both start at the beginning of development when pluripotent stem cells differentiate into cells with specialized form and function. These data suggest that aging involves developmental processes that start in early development and continue through the lifespan. Indeed, many aspects of development such as cell division, differentiation, and cell senescence do not end at the onset of adulthood. Various degrees of turnover occur in most systems ranging from high in blood, skin, intestine, and bone, for example, compared to low turnover in the brain and heart. These processes are necessarily slowed down with the onset of adulthood as they now serve to maintain and repair rather than construct new organs. However, eventually, the repair processes cannot keep up with ongoing damage leading to loss of homeostasis and tissue integrity associated with aging. The DNAm clock starts ticking and is accelerated during development and may follow a logarithmic function [61,62]. It seems reasonable to assume that the rate at which the DNAm clock ticks through adulthood would correlate with the relative rate of aging within and among various species. For example, recently, mouse DNAm clocks have been shown to tick faster than the human clock reflecting their shorter lifespan [63,64,65]. There is indeed a wide variation in the rate of aging among and within a species [6]. In addition, there are wide variations in life history, with some organisms having an extended juvenile period and short adulthood and others having the opposite life history. Thus, it would also be interesting to measure DNAm aging across species with varying lifespan and life histories. Toward this end, Horvath et al. have measured DNAm across 128 mammalian species in multiple tissues at various ages in an effort to develop a universal mammalian DNAm clock [66]. Epigenetic programming may also impact the extremely wide variation in lifespan within a species, for example, in social insects such as bees and termites [67].

The DNAm clock appears to correlate well with the plasticity of lifespan. Multiple DNAm clocks show accelerated aging in many disease conditions, including cancer, and they can successfully predict all-cause mortality and frailty [47,68,69]. Accelerated epigenetic aging in cancer cells seems contradictory to their escape from senescence. However, hTERT immortalized cells also continue to age in culture by the Horvath DNAm clock [70], indicating that although epigenetic aging and replicative senescence are both linked to the rate of aging, they can become uncoupled under certain conditions [71]. In this way, abnormal cancer cells differ from the germline, which escapes both replicative and epigenetic aging. The Horvath DNAm clock shows age acceleration in diseases that are associated with premature aging and accelerated telomere attrition, such as Werner’s syndrome and chronic HIV infection [72,73]. The newer blood and skin clock developed by Horvath measures accelerated aging in samples from Hutchinson–Gilford progeria [60]. DNAm clocks are able to measure age deceleration as well, for example, in humans undergoing treatments such as exercise, nutritional interventions, and other lifestyle factors associated with increased life expectancy [74,75]. Moreover, decelerated DNAm aging is observed in calorie-restricted and long-lived dwarf mice [63,64]. DNAm age deceleration is also seen in extremely long-lived humans who also have extended healthspans [76]. Thus, DNAm clocks are useful for measuring both accelerated and decelerated aging within a species. Finally, the third type of aging clock we will discuss involves repetitive DNA elements, including ribosomal DNA (rDNA), transposable elements (TE), and retroviral transposable elements (RTE) that are dispersed throughout the genome.

## 4. Nature of Somatic Cellular Clocks—Transposable Element Clock

The TE clock measures changes in expression and mobilization of transposable elements, a type of repetitive DNA sequence in the genome. One of the earliest observations of a connection between DNA repetitive elements and aging was made in yeast, where accumulation of extrachromosomal ribosomal (rDNA) circles (ERC) was identified as a cause of aging [77]. Yeast undergoes aging even though they are single-cell organisms because the yeast cell acts as both soma and germline. The mother cell gives birth to daughter cells by a budding process that partitions the aging factors (e.g., rDNA circles) asymmetrically such that they remain in the mother cell, which ages while the daughter cells are rejuvenated. The rDNA locus is a large family of gene repeats associated with homologous recombination and ERC. The chromatin modifier, Sir2 (an NAD+ dependent class III histone deacetylase), which promotes chromatin silencing at yeast mating-type loci and telomeres, was found to stabilize the rDNA loci [78]. Moreover, the loss of Sir2 accelerates replicative aging in yeast, and its overexpression leads to extended lifespan [79]. These observations led Oberdoerffer and Sinclair to propose the relocation of chromatin modifiers (RCM) hypothesis, which proposes that chromatin-modifying factors, whose silencing activity normally maintains cellular identity, are relocated to sites of DNA damage for repair. The chromatin modifying factors then quickly return to maintain silencing in young organisms, but in aging organisms, the chromatin modifying factors cannot keep up with increasing DNA damage leading to a loss of their normal silencing function and consequent loss of cellular identity and dysfunction associated with aging [80]. They speculate that RCM is an ancient survival mechanism that coordinates DNA repair and activation of survival genes in order to, for example, prevent mating while DNA damage is present [45,80]. A similar mechanism may be present in mammals as related sirtuins, SIRT1 and SIRT6, relocate from diverse gene promoters and repetitive DNA to sites of DNA damage to facilitate repair [81,82,83]. As a consequence, transcription of repetitive sequences and age-related genes occurs but can be prevented by overexpression of SIRT1 or SIRT6 [81,83]. Further evidence for conservation of RCM in mammals includes the observation that SIRT7 stabilizes rDNA in mice via recruitment of SIRT1 and DNMT1 [84]. In addition, Peredes et al. have demonstrated that sirtuins prevent senescence in human cells by stabilizing rDNA [85]. Taken together, these studies point to a critical role of repetitive DNA sequence silencing in the regulation of aging.

Although there is evidence for the association of changes in rDNA as well as nucleolar size and activity with aging and senescence [86,87], it was only recently that an rDNA methylation clock had been proposed. This is likely due to the exclusion of rDNA repetitive sequences in genomic builds and in methylation arrays used in public databases. Wang and Lemos have developed an rDNA aging clock based on whole-genome bisulfite sequencing data that revealed sites of age-related CpG hypermethylation in rDNA relative to the remaining genome [88]. The clock accurately estimates the age of individuals within a species, and because of the ultra-high conservation of the rDNA sequence and CpG methylation sites, it can be used across species as diverse as human, mouse, and dog [88]. Importantly, the ribosomal DNA methylation (rDNAm) clock age is low in human embryonic stem cells (hESC), which is consistent with a resetting of all aging clocks at each generation in the immortal germline. Moreover, the flexibility of the clock is demonstrated by its deceleration in response to well-established aging interventions such as caloric restriction and mutation of growth hormone receptor (GhR) in mice [88]. Thus, the rDNAm clock may act as a universal age marker for higher organisms and could be useful for estimating the age of animal species in the wild for which chronological age data is not available. Both site-specific hypermethylation in the form of the rDNAm clock and hypomethylation, which activates rDNA transcription and double-stranded breaks (DSB), appear to play a role in aging in higher organisms. However, more studies are needed to connect the rDNAm clock with observations of the role of rDNA in aging in lower animals such as *Caenorhabditis elegans* and *Drosophilia melanogaster* where very little or no CpG methylation is detected.

The observation of decreased heterochromatinization of repetitive elements including TE and increased TE transcription and mobilization with aging supports the hypothesis of a TE clock [89,90,91]. Heterochromatinization is promoted by DNAm via methyl CpG binding protein (MeCP2), which recruits histone deacetylases and other chromatin regulators. Overall, DNA methylation at repetitive sequences decreases with aging, suggesting a relationship with the DNAm clock [89]. TEs are ubiquitous and exist in the genomes of virtually all organisms, playing an important role in genome regulation and evolution [92,93]. Remarkably, about half of mammalian genomes are made up of these descendants of ancestral viruses [94]. The universality of TEs suggests that they may be fundamental to common biological processes such as development, aging, and evolution. Indeed, the transient activation of certain TEs in early development may have profound effects on gene expression patterns [95]. TEs are classified as DNA transposons that use a transposase-dependent cut and paste transposition mechanism and retrotransposons (RTEs) that use a reverse transcriptase-based copy and paste mechanism [96]. RTEs make up about 40% of mammalian genomes. They consist of endogenous retroviruses (ERVs) that contain long terminal repeats (LTRs) and non-LTR RTEs consisting of the long-interspersed elements (LINEs) and short-interspersed elements (SINEs) [97]. ERV and LINE transpositions are autonomous, but SINEs can be activated in trans by LINE reverse transcriptase. The vast majority of RTEs are inactive except for evolutionarily recent RTEs such as LINE 1 (L1) and the HML-2 group, which are activated in early embryogenesis, neurogenesis, and certain cancers [98,99,100,101]. The HML-2 provirus, a subtype of human endogenous retrovirus-K (HERV-K), can even form viral-like particles in teratocarcinoma cells, melanoma cells, blastocysts, and hESCs [98,102]. In early embryos, activation of HERV-K may be an evolutionary defense against viral infection via the interferon-induced transmembrane protein-1 response [102]. TE mobilization is associated with DSBs, which are in turn associated with aging [103,104,105]. Indeed, Wood et al. identified an age-dependent increase in RTE mobilization in *Drosophila* as well as a correlation of a reduced rate of mobilization with lifespan extension by caloric restriction [106]. Furthermore, interventions that increase maintenance of heterochromatic repression were also found to suppress TE mobilization and increase lifespan [106]. The effects of increased RTE mobilization were suppressible using the reverse transcriptase inhibitor lamivudine [106]. Similar associations of TE activation/mobilization and aging have been found in mammals [91]. L1 copy number increases with the onset of senescence in cultured fibroblasts, and an increase in mobilized RTEs has been found in late-stage senescent cells both in vitro and in vivo [90,107,108]. Moreover, somatic L1 elements are derepressed in SIRT6 knockout mice, which exhibit a progeroid rapid aging phenotype [109]. The accumulation of L1 cDNA was found to activate an interferon response resulting in sterile inflammation, a hallmark of aging, in these mice, suggesting a mechanism by which RTE activation can contribute to organismal aging [109]. Reverse transcriptase inhibitors improved the healthspan and extended lifespan of the SIRT6 KO mice providing further evidence of the role of TE activation in aging and potential treatments for age-related disease [109]. Further evidence for the role of TEs as an aging clock in somatic tissues comes from a recent studying revealing that repetitive element transcription can be used as an age predictor in diverse species, including mice and humans [110]. Although further studies are needed to elucidate, TE transcription and mobilization leading to DNA damage and inflammation clearly play an important role in the aging of time-bound somatic tissues.

Non-coding RNA (ncRNA), including microRNA and P-element-induced wimpy testes (PIWI) interacting RNA (piRNA) may play an important role in regulating the rate of TE-associated aging by participating in the suppression of TEs by gene silencing and transcript/translational regulation [111,112]. Accordingly, the PIWI-piRNA pathway has been proposed as a mechanism for immortal cell regeneration as well as regulating the rate of aging [113,114]. The piRNA is processed from long ncRNAs that are enriched in TE sequences and are used to guide PIWI proteins to complementary RNAs to silence the TE [115]. The PIWI-piRNA pathway also plays a role in histone modification and chromatin silencing [115,116]. Accordingly, the clock-free immortal germline largely escapes TE mobilization due to active suppression of these elements by the PIWI-piRNA pathway, although limited transposition may play a role as a driver of genetic diversity and species evolution [115,117]. The PIWI-piRNA pathway is also active in somatic stem cells with high proliferative potential and in cells that maintain and regenerate tissue in highly regenerative lower metazoans that often display negligible senescence. For example, PIWI-pathway is found in the archeocytes of sponges [118], neoblasts in planarians [119], and somatic stem cells of hydra [120]. Recently, a direct association of PIWI-piRNA and TE repression was identified in hydra [121]. Studies have also identified piRNA in the regenerating limb of the axolotl, presumably acting as a silencing response to activation of L1 and other RTE during a reprogramming-like activation of regenerative cells of the blastema [122]. PIWI protein expression in somatic cells occurs primarily in stem/progenitor cells and is species-specific [113]. In humans, PIWI proteins are expressed in CD34+ hematopoietic progenitor cells [123]. A dramatic example of the influence of TE activation and PIWI-piRNA on the regulation of lifespan is seen in the termite species *Macrotermis bellicosus*, where the reproductives (king and queen) have maximum lifespans of at least 20 years compared to several weeks for non-reproductive workers [124]. This study identified silencing of TEs in both young and old long-lived reproductives while down-regulation of PIWI-piRNA pathway correlated with active TE transcription in old but not young short-lived worker termites [124]. The termite colony can be viewed as one superorganism where the reproductives represent the germline and the workers the soma. Thus, the TE clock provides an intrinsic mechanism that may account for an adaptive stretching or contraction of lifespan.

## 5. Nature of Somatic Cellular Clocks: Aging and Development

The three clocks we have described that bind the soma to time can be thought of as part of a larger developmental clock. We have previously described, in our somatic restriction hypothesis, a series of developmental steps whereby there is a progressive loss of the replicative and regenerative potential of somatic cells and tissues that eventually leads to age-associated disease and tissue degeneration (Figure 2) [8]. In accordance with Williams’ antagonistic pleiotropy hypothesis, this restriction is adaptive in early life because it prevents unregulated growth such as malignant cancer cells but is deleterious in later life, leading to fibrotic healing and aging. As we have discussed, the first step in somatic restriction is the loss of immortal replicative potential at the pluripotency transition (PT), after which the telomeric clock (loss of telomerase) and DNAm clock begin to tick. The next developmental phase is the embryonic period, where all major body structures are formed, and regenerative capacity is maximal. This stage ends with the embryonic to fetal transition (EFT) followed by a period of rapid growth with some residual regenerative capacity that is further diminished following the neonatal transition (NT). Finally, there is the period from birth to adulthood ending the growth stage with the adult transition (AT). Each transition is defined by a characteristic shift in gene expression pattern. For example, we identified the expression of certain genes such as the protocadherin, *PCDHB2*, as pre-EFT markers, and other genes such as the mitochondrial complex IV gene, *COX7A1*, as post-EFT markers [125]. Indeed, we have observed a shift in expression of the protocadherin cluster from the α and β forms during embryogenesis to the ɤ forms in the fetal/adult state, with this pattern reverting to the embryonic state in cancer cells [126]. We propose a model where topological changes in chromatin mediated by the shift from *LMNB1* to *LMNA* expression drives the shift in protocadherin gene expression at the EFT [126]. The rapid growth of the embryo and fetus is markedly curtailed, and there is a loss of residual regeneration, such as in the heart shortly after the NT, which is marked by a change in gene expression at many imprinted loci such as loss of insulin-like growth factor-2 expression. Following the cessation of growth at the AT, the somatic clocks continue and there is a progressive loss of tissue integrity owing to decreased ability to accurately regenerate cells and tissues leading to the increased senescence (decreased survival rate) we associate with aging.

Interestingly, the DNAm clock ticks faster following a logarithmic relationship up to the AT and linear relationship to chronological age through adulthood [47]. The telomere clock also ticks more rapidly in early development, with half of the telomere length lost before birth. Data from leukocyte telomere length indicate rapid telomere attrition up to age 3, with the rate reduced through childhood and slowing down again after the AT [127]. Horvath and Raj have speculated on the link between epigenetic aging and development, noting an overrepresentation of clock methylation sites near genes that are regulated by the Polycomb repressive complex (PRC), which plays an important role in embryonic development, stem cell differentiation, and tissue homeostasis [61]. The proposal that aging is a consequence of development was put forth in 1907 by Minot, who wrote that “it is during the embryonic period that the loss of power of growth is greatest” and that “the condition of old age is merely a culmination of changes which have been going on from the first stage of the germ up to the adult” (1). Minot’s words are consistent with an accelerated DNAm clock during development and with the loss of regenerative capacity at the EFT as described in our somatic restriction hypothesis [8]. More recent models and data supporting aging as a consequence of development have been reviewed by Magalhaes and Church [128]. Indeed, there appears to be an impact of the early developmental environment and aging rate/longevity, as indicated by studies in mice and humans [129].

Heterochronic genes such as *LIN28* (a pre-EFT marker and reprogramming factor) alter the timing of developmental transitions and thus potentially impact regenerative capacity and lifespan. Metamorphosis is an example of developmental processes occurring at various times during the life history as it always involves extensive remodeling of organs and tissues at the histological level, including the growth of new organs and limbs [130]. Perhaps one of the most remarkable examples of heterochrony is the variation in the timing of metamorphosis at various stages of the life history or not at all in the case of paedomorphism, where an organism retains juvenile characteristics throughout adulthood [130]. The paedomorphic axolotl, for example, rarely undergoes metamorphosis, thus retaining its juvenile aquatic form throughout adulthood. When it does metamorphose under dry conditions, the resulting salamander lifespan is greatly reduced [131]. The heterochronic gene, *Lin28*, is expressed in the regenerating limb of the axolotl, and mice engineered to express *Lin28* in adulthood have enhanced regenerative capacity [132]. Mammals such as the naked mole-rat and humans similarly may exhibit neoteny, where embryonic and juvenile physiological, metabolic, and gene expression characteristics of closely related species are retained through adulthood [131]. Retention of these early developmental features may account for their relatively long lifespans. Indeed, the naked mole-rat, which lives about 10-fold longer than a mouse, may be the first mammal reported to exhibit negligible senescence [133]. Humans have about a 4-fold greater maximum lifespan than chimpanzees. In humans, fetal gene expression patterns in the dorsolateral prefrontal cortex that drop rapidly after birth in the chimpanzee are delayed, peaking in childhood and persisting through adolescence [131]. Accordingly, the DNAm clock has recently been shown to tick faster in chimpanzees than in humans, which is consistent with their shorter lifespan [134]. Thus, naked mole-rats and humans are examples of delayed development being associated with lengthened lifespan. The recently developed universal mammalian DNAm clock also correlates the rate of development to sexual maturity to maximal lifespan across mammalian species with lifespans varying from 3 to over 100 years [66]. Further studies are needed to determine how the rate of the DNAm clock during development and the timing of developmental transitions impact longevity and aging.

## 6. Uncoupling Biological Time from Chronological Time

In the late 19th century, Weismann accurately described the dual nature of life, separating it into the immortal germline and mortal soma. Here we have described three cellular clocks and their relation to the limited lifespan of somatic cells. Certainly, much more likely exist, such as transcriptomic, proteomic, and metabolomic clocks [135,136], but the three discussed here are likely primary because they directly affect the genome either by telomere shortening, DSB, or chemical modification. The molecular basis of the immortal germline extending from the first unicellular organisms through to the current multitude of lifeforms would not be understood until the mid-20th century with the discovery of DNA. Although the information stored in DNA changes through mutation and reassortment of genetic alleles as a driver of evolution, the DNA nevertheless retains all the information needed to make a new body, so each generation is born young. As Weismann expressed in a letter to Nature, “An immortal unalterable living substance does not exist, but only immortal *forms of activity* of organized matter” [137]. We may now extend this to “immortal *information encoded in DNA* of organized matter.” As we have described, the germline either escapes or resets the aging clocks so that the genomic information is preserved. Thus, the germline preserves the information needed to perpetuate itself indefinitely and to make a new body, even though the body that carries it ages.

Unlike the immortal germline, the aging soma follows a predictable course that is coupled to time yet flexible enough to accelerate or decelerate depending on environmental and evolutionary pressure. Remarkably, it is now possible to reverse the cellular clocks of development and aging, winding them back from an aged somatic cell to the earliest stage of life. We have evidence from cloning (somatic cell nuclear transfer (SCNT)) experiments first performed in frogs and later in mammals, including humans [138,139,140] that while somatic cells age and accumulate the key hallmarks of aging with time [11], the information to make a young cell is preserved in the DNA. During SCNT, exposure of a somatic nucleus to germline factors in the egg resets the aging clock to zero, such that cloned animals are born young as if they were conceived from two germ cells [141,142,143]. Moreover, Yamanaka demonstrated over a decade ago that forced expression of as few as four germline factors, OCT4, KLF4, SOX2, and c-MYC (OKSM), alone could be used to reprogram development in vitro, reverting a differentiated cell back to an induced pluripotent stem cell (iPSC) [144]. However, it was initially unclear whether the reprogramming of a differentiated cell would also result in reverting the hallmarks of aging, including telomere shortening and DNAm age. Initial studies suggested that iPSC lines were potentially limited in their differentiation capacity because of short telomere [145,146]. Determination of telomere resetting in iPSC lines is complicated by wide variation in original embryonic telomere length. We overcame this issue by studying telomere resetting in an hESC-derived fibroblast cell line such that the original hESC telomere length could be easily established [147]. The results showed that telomere length can be restored to the length of the parental hESC line by reprogramming [147]. Additional studies have demonstrated that reprogramming resets telomere length in cultured senescent fibroblasts and even in fibroblasts obtained from centenarian donors [148,149]. Indeed, we recently demonstrated that reprogramming could reset telomere length even in cells from a donor near the current limit of human aging (114 years) [150]. As mentioned, the telomere clock ticks rapidly in patients with premature aging syndromes, but we and others have shown that telomeres in these cells can also be reset to embryonic length by reprogramming [150,151,152]. In addition, epigenetic markers of aging such as senescence-associated heterochromatin foci (SAHF) and heterochromatin protein-1α (HP1α) are restored to young levels by reprogramming cells from prematurely aged as well as naturally aged donors [148,151]. Other hallmarks of aging, including mitochondrial fitness, have been shown to be reversed by reprogramming [148,153]. Derivatives of reprogrammed iPSC have also been shown to retain a young phenotype. Accordingly, differentiation of iPSC lines to their respective cell type of origin, such as mesenchymal stromal cells, hematopoietic stem cells, and fibroblasts, results in a rejuvenated phenotype as assessed by transcriptome and epigenome analysis [150,154,155]. Moreover, iPSC from old mice can be differentiated, in vivo, into heart cells that closely resemble their younger counterparts [156]. Taken together, these studies strongly support the reversal of both developmental stage and aging by germline factor reprogramming to iPSCs.

Interestingly, examples of reversal of developmental stage can be found in nature as an adaptation to environmental conditions as observed in the expression of embryonic pathways in the regenerating limbs of urodeles or in oncogenic transformation. One of the most profound natural reversals, noted by Weismann in 1883, is seen in the hydrozoan, *Torritopsis dohrnii*, which normally develops into a medusa from a benthic polyp form but undergoes a reversal from medusa to polyp via transformation to a cyst intermediate under adverse conditions such as injury, starvation or senescence [157]. Comparative transcriptome analysis of the cyst versus medusa and polyp forms shows enrichment in gene pathways such as DNA integration, transposition, repair, and telomere maintenance suggesting germline-like maintenance of genome integrity in the intermediate cyst stage [158]. However, it remains to be determined whether there is suppression of TEs at the cyst stage, as in the hydra and long-lived termite reproductives. In contrast, somatic cell-associated pathways such as cell signaling, aging, and differentiation were downregulated in the cyst stage [158]. Another example of developmental reversal occurs in the process of sex transition in a marine fish, the bluehead wrasse, which remarkably involves some of the same germline factors used for cell reprogramming. The female wrasse undergoes a rapid behavioral and complete physical transformation to sperm-producing terminal phase (TP) male in 8–10 days in response to the loss of the dominant TP male and the subsequent increase in cortisol levels. Mutually antagonistic male and female gene networks determine and maintain gonadal fate in fishes and thus account for the retention of a feature of embryonic development, bipotentiality of sex, into adulthood. The mechanism of transformation in the bluehead wrasse has been recently elucidated at the molecular level [159]. Transcriptomic and methylomic analysis of gonads at various stages from the female through TP male reveals that the gonads pass through an undifferentiated midway stage that resembles mammalian pluripotent stem cells (PSC) and primordial germ cells (PGC) rather than transdifferentiation from the female to male state [159]. For example, Polycomb group members of PRC2, which are responsible for tri-methylation of lysine 27 on histone H3 (H3K27me3) and are downregulated in PSC, are also downregulated during the midway stage of the female to male transition. The variant histone H2A.2, which is also low in PSC, shows a similar pattern. In addition, writers and erasers of histone acetylation are expressed dynamically during the transition. Evidence of extensive reprogramming of DNA methylation was also observed with upregulation of ten-eleven translocation demethylase (TET) (as in PSC and PGC) midway through transition leading to a shift from female to male pattern of methyltransferases. Genome-wide DNA methylation changes from the female to male pattern were found to occur. Notably, like the transcriptome, the midway methylome state represents a developmental shift rather than an intermediate differentiated state. Indeed, it would be interesting to determine whether the DNAm age of the gonad regresses during the transition if an epigenetic aging clock can be created for marine fishes. These data illustrate remarkable plasticity in a normally committed developmental process of sex determination via epigenetic reprogramming involving transition through an earlier developmental state. Remarkably mutually antagonistic gene networks that both determine and maintain sex in adulthood have also been found in mice indicating conservation of some degree of plasticity even in mammals [160,161]. Given the relationship of aging to developmental processes and examples of the maintenance of developmental plasticity into adulthood, could epigenetic reprogramming uncover latent phenotypic plasticity in mammals to return aging adults to a younger epigenetic state and thereby reverse aging?

Given the astonishing degree of phenotypic plasticity observed in the forward and reverse programming of development and aging in nature and in the laboratory, we may perhaps view young and old organisms as two epigenetic states of the same genome with the young state contained (as information) within the old and the old state contained in the young as a potentiality that becomes actualized by the ticking of the cellular clocks with the passage of time (Figure 3). If aging is indeed a continuation of development as suggested by the behavior of cellular aging clocks, then it is not surprising that reprogramming can reverse the forward programming of both development and aging. However, it also raises the intriguing possibility of reversing aging without loss of developmental status if reprogramming is capable of turning back development and aging in the reverse order in which they occurred. A study of transcriptomic and methylomic data from a time course of fibroblasts undergoing reprogramming suggests that this may indeed be the case and that reprogramming is reverse programming of aging first followed by a reversal of developmental state (Figure 4) [162]. The study shows a linear decrease in DNAm age during the early phase of reprogramming (partial reprogramming; day 3–11) when fibroblast identity has not been lost. At this early stage, the cells undergoing reprogramming have a high propensity for spontaneous reversion to their initial fully differentiated state. Importantly, early and late markers of pluripotency (*LIN28*, *DNMT3A*, *ZIC3*, *TERT*) are not induced, and the fibroblast defining genes are not fully repressed during the initial age reversal/partial reprogramming phase before the DNAm age has reached zero (day 20). Similarly, genes we have identified as fetal/adult-specific (post-EFT), such as *COX7A1* and *ADIRF*, are reduced but not to minimal levels during partial reprogramming, and embryonic-specific genes (pre-EFT) such as *PCDHB2* are not expressed until day 20 (Figure 4) [125]. Another study has recently reported rejuvenation of transcriptomic and DNAm age by as much as 30 years in transiently reprogrammed fibroblasts from middle-aged donors [163]. It will be important to obtain single-cell analysis to determine the dynamics of these changes within subpopulations of the reprogramming factor-treated cells. The data suggest, however, that reprogramming first runs the epigenetic aging steps in the reverse order that they occurred during the life history followed by a reversal of developmental state. Therefore, these data indicate the potential to apply reprogramming methods therapeutically for rejuvenating aged cells, tissues, and perhaps whole organisms without loss of cell identity.

Initial attempts at in vivo reprogramming using 4-factor (OKSM) reprogramming were not encouraging because they resulted in extensive tumor formation [164,165]. However, Ocampo et al. later reported reversal of epigenetic aging markers with no signs of tumor development in transgenic OKSM mice when the factors are induced at a lower dose and on a cyclic schedule of 2 days on and 5 days off [166]. Instead of tumor formation, cyclic OKSM induction led to amelioration of the premature aging phenotype and extension of lifespan in a mouse model of HGPS. They further demonstrated that cyclic OKSM induction resulted in greater resistance to metabolic disease following pancreatic injury and increased repair of skeletal muscle damage in physiologically aged mice. Additional in vivo studies have shown reduced scar formation of skin wounds using viral vector-mediated partial reprogramming and protection against liver damage using a small molecule approach [167,168]. The question remained, however, as to whether partial reprogramming in vivo reversed the DNAm clock. Two recent studies provide evidence suggesting that a reversal of the DNAm clock during in vivo reprogramming may be possible [169,170]. In one study, transient expression of six reprogramming factors, OCT4, SOX2, KLF4, LIN28, c-MYC, and NANOG (OSKLMN), was obtained using mRNA transfection, which resulted in a reversion of fibroblasts and endothelial cells to a younger phenotype as measured by epigenomic markers (tri-methylation of lysine 9 on histone 3 (H3K9me3), HP1γ, and lamina-associated polypeptide 2α (LAP2α)), mitochondrial fitness, autophagy, transcriptome, and DNAm clock [170]. Transient expression of OSKLMN mitigated the inflammatory phenotype of osteoarthritic chondrocytes in culture, and implants of ex vivo OSKLMN treated aged mouse or human skeletal muscle stem cells resulted in rejuvenated muscle regenerative response following injury in physiologically aged mice with no signs of tumor formation [170]. In another in vivo study, only three factors (OKS) were delivered to mice using an inducible adeno-associated virus vector (AAV). The results indicate that induction of OKS resulted in the protection of retinal ganglia cells and regeneration of axons in an optic nerve crush injury model, and the same treatment restored vision in a mouse model of glaucoma [169]. Importantly, the DNAm age was reversed by the treatment in both models, and the regenerative effects were Tet1 and Tet2 dependent in the optic nerve model, indicating that these reprogramming factors may be “reverse programming” the epigenetic clock to an earlier state (DNAm age) with an enhanced regenerative capacity [169]. The introduction of germline factors into somatic cells in vivo reveals a potential for phenotypic plasticity in adult mammals that was previously observed only in lower life forms.

The prospect of reversing organismic aging could potentially reduce or eliminate many age-associated degenerative diseases such as heart disease, cancer, diabetes, osteoporosis, sarcopenia, neurodegenerative disease, and skin aging. Thus, rejuvenation therapies could potentially alleviate many of the social and economic costs that we face globally because of an ever-increasing demographic shift to an older population. However, critical barriers remain to develop rejuvenating therapies that involve germline factor reprogramming. For example, more studies are needed on naturally aged experimental animals to determine the effect of such treatments on lifespan and health span. Moreover, careful control of the timing and dosage of reprogramming factors will need to be developed to minimize the risk of tumor formation. The further development of epigenetic aging clocks in experimental animals and tissue culture will be helpful for screening reprogramming agents and delivery strategies. Strategies for transient expression of reprogramming factors include AAV mediated inducible gene delivery, RNA transfer, and small molecules [168,169,170]. The use of extracellular vesicles such as exosomes to deliver reprogramming RNAs is also a promising approach because of their long half-life in vivo and low immunogenicity [171]. Pre-clinical studies have shown the efficacy of engineered exosomes for the treatment of pancreatic cancer, and they can be effectively scaled using standard bioreactors [172]. In addition to their potential as a delivery vehicle, exosomes produced by young mesenchymal stem cells (MSC) may, on their own, be an effective way to slow or reverse some aspects of aging [173]. The development of reprogramming therapies may even benefit from the considerable progress in RNA delivery and manufacturing that has resulted from the accelerated efforts to develop COVID-19 vaccines [174]. Careful attention will be needed to assess the appropriate indications and regulatory considerations when developing clinical studies. Initial proof of concept studies may involve specific indications such as arthritic joint disease; however, later studies may be possible using resistance to age-related disease as an outcome as in the proposed TAME study [175].

## 7. Concluding Remarks

In this review, we have described three cellular clocks that bind somatic cells to time and compared this with the immortal germline, which in its capacity for indefinite renewal, is uncoupled from time. We have also discussed the plasticity of the aging clocks that tick faster in accelerated aging or slower in decelerated aging. The initiation of the DNAm aging clock, the telomeric clock, and perhaps other clocks at the beginning of development suggests an intimate relationship between development and aging. Indeed, the adaptation of developmental clock rate to environmental pressure could account for the wide variation in lifespan observed between species. For example, humans and mole-rats exhibit neoteny, where slowing the rate of development correlates with an extension of lifespan. The application of germline strategies in somatic stem cells has resulted in the remarkable regenerative capacity of lower life forms that are capable of indefinite lifespans, such as sponges, planarians, and hydra. This regenerative capacity has become increasingly restricted as more complex life forms evolved, being confined to the pre-EFT period in mammals. However, retention of extensive capacity for regeneration is observed in lower vertebrates, including fishes, amphibians, and reptiles, which also exhibit remarkable phenotypic plasticity in their capacity for metamorphosis and in certain cases of remarkable reversals of developmental stage and sexual development. Finally, reprogramming using germline factors can uncover a similar but latent phenotypic plasticity in mammals by reverting both the developmental state and cellular age. Indeed, both natural phenotypic plasticity in the blue wrasse and partial reprogramming involve the repression of DNA methyl transferases (DNMT) and induction of demethylases (TET) which, by a yet-to-be-determined mechanism, may enable the DNAm clock to tick backward. The discovery that partial reprogramming can reverse the aging clock without permanent alteration of cellular identity has led to initial studies that demonstrate the potential to reverse organismic aging. Although there are many challenges ahead, our current understanding of cellular clocks and our ability to reprogram them using germline factors opens the door to many promising therapeutic approaches to slowing down, preventing, or reversing aging itself and thus treating the many age-related diseases that burden society. Indeed, if these approaches can be made practical and scalable, we may find ourselves in a future in which we have no time to age.

## Figures and Tables

**Figure 1 genes-12-00611-f001:**
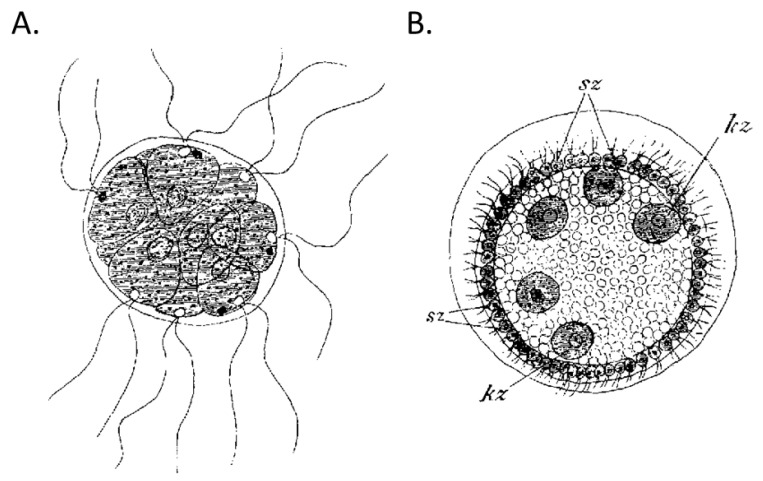
The appearance of germline and somatic cells in the evolution of metazoans. Two Volvocinian genera that were used by Weismann to illustrate the difference between (**A**) a homoplastid organism (*P. morum*) consisting of a single cell type and the evolution of (**B**) a heteroplastid (*V. minor*), which has evolved to consist of two primary cell types: the germline cells (Kz) that form the internal embryos and somatic cells (Sz), which are subject to death upon which the developing embryos are released.

**Figure 2 genes-12-00611-f002:**
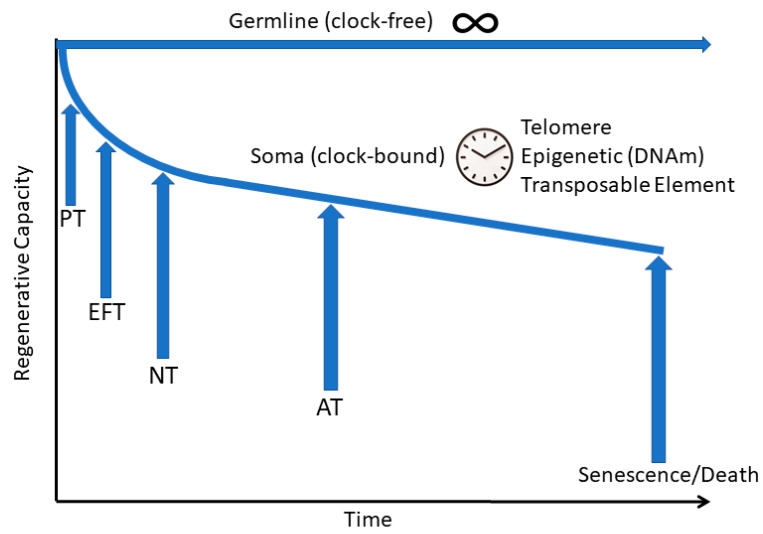
Somatic restriction model of regenerative capacity in germline versus somatic cells. The immortal germline is capable of indefinite replication (clock-free). In contrast, the cells of the body are clock-bound and therefore lose regenerative capacity as the cellular clocks (telomere, DNAm, and transposable element) tick forward soon after development starts and continue through a series of distinct transitions marked by changes in gene expression and epigenetic configuration of chromatin. PT, pluripotency transition; EFT, embryonic to fetal transition; NT, neonatal transition; AT, adult transition.

**Figure 3 genes-12-00611-f003:**
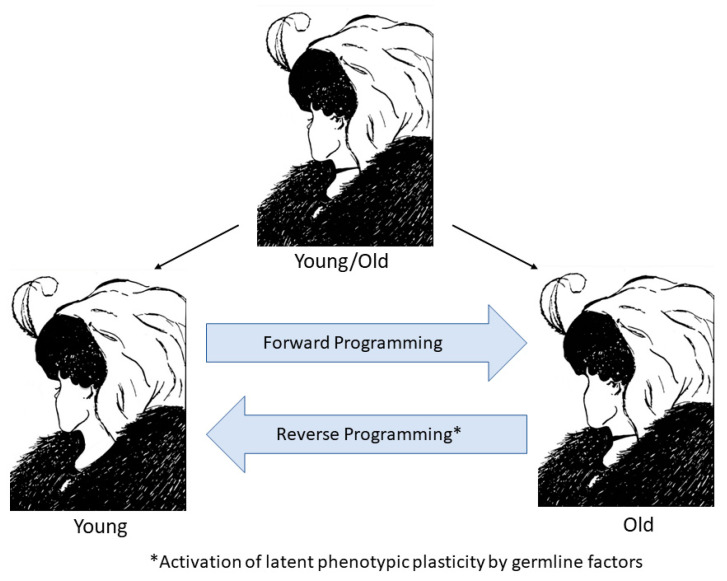
The two faces of epigenetic aging. The top illustration is an illusion designed so that the viewer will see either a young woman or an old woman. However, both are present in the same figure, just as the young and old woman are present in the same person as the genome’s epigenetic potential. Forward programming takes the young woman through a series of epigenetic states to an old woman as the DNAm, and other cellular clocks tick forward. Reverse programming can potentially be achieved using germline factors to revert the genome to an earlier epigenetic age.

**Figure 4 genes-12-00611-f004:**
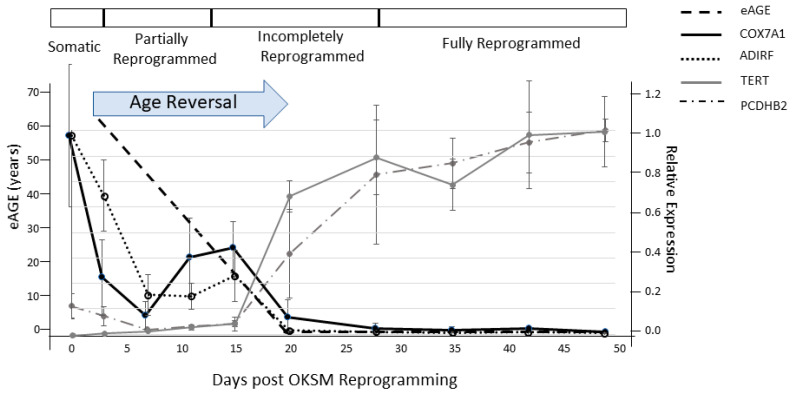
Age reversal precedes reversion of developmental state during reprogramming of fibroblasts using OKSM germline factors. Post-EFT genes (COX7A1, ADIRF) are rapidly reduced during early (partial) reprogramming (day 3–20) while a linear decrease in DNAm age is occurring. Pre-EFT genes are turned on at day 20 when DNAm age has gone to zero, and their expression increases as cells reverse their developmental state to full pluripotency (≥28 days). The data suggest the possibility of age reprogramming without permanent loss of cell identity. Figure is adapted from Olova et al. Figure 1 [162].

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
