# Peer review of "No Time to Age: Uncoupling Aging from Chronological Time"

_genes, 2021, doi:10.3390/genes12050611_

Round 1
Reviewer 1 Report
The review by Larocca et al. entitled "No time to Age: Uncoupling Aging from Chronological Time" summarizes a complex bulk of information that biologists and aging scientists have been discussing for several decades. In addition, they explore the reprogramming of cells as a mean to reverse the tick of the biological aging clock. the authors start exploring differences between different organisms lifespan and germline vs somatic cells. They explore three biological clocks (telomere, DNA methylation and transposable elements). This is a very comprehensive manuscript, I have only some minor comments.
Comments:
Lines 35-37: Please add a reference for the wear and tear mechanisms.
Lines 145-148: Although introduced later you can include the references for HIV and progeria earlier.
Lines 166-167: Please introduce the Hayflick limit theory. Her it is disconnected.
Line 180-181: Do you mean "healthy diets"? could you be more specific?
Lines 205-206: There are several gestational clocks using placenta and cord blood with different levels of accuracy including Bohlin et al., and Knight et al. You can mention those here too including their potential problems (Simpkin et al.). Other markers such as the Fetal cell of origin (FCO, Salas et al.) have shown other markers of the transition between the fetal to adult markers.
Lines 243-258: One piece that is missing here is the mitotic clocks (epiTOC and epiTOC2, Teschendorff et al), which are important to clarify the somatic ticking of the clocks.
Line 292: Do you mean bisulfite?
Author Response
We are grateful for the careful reading of the manuscript and well thought out suggestions for minor revisions. We believe we have addressed the specific points and that the manuscript has been significantly improved upon further editing in response to the review. We have responded to the reviewer’s points as follows (Lines from reviewer’s comments refer to original, lines in our response refer to the revised manuscript):
Reviewer 1:
- Lines 35-37: Please add a reference for the wear and tear mechanisms. Line 37: We have added 2 references describing entropic theories of aging by Harman and Hayflick.
- Lines 145-148: Although, introduced later you can include the references for HIV and progeria earlier. Line 150-151: We have included the 2 references at this earlier point in the revised manuscript.
- Lines 166-167: Please introduce the Hayflick limit theory. Here it is disconnected. Lines 109 and Line 169: The Hayflick limit is introduced earlier, now in the manuscript 2 paragraphs prior.
- Line 180-181: Do you mean “healthy diets” could you be more specific? Line 182: The word diet has been replaced by calorie restriction to make it more specific to the work described in the references.
- Lines 205-206: There are several gestational clocks using placenta and cord blood with different levels of accuracy including Bohlin et al., and Knight et al. You can mention those here too including their potential problems (Simpkin et al.). Other markers such as the Fetal cell of origin (FCO, Salas et al.) have shown other markers of the transition between the fetal to adult markers. Lines 210 and 211: All of the suggested additional references have been added.
- Lines 243-258: One piece that is missing here is the mitotic clocks (epiTOC and epiTOC2, Teschendorff et al), which are important to clarify the somatic ticking of the clocks. The epiTOC (epigenetic timing of cancer) clocks measure cancer risk by estimating replicative burden. However, while these are important observations, unlike the other clocks we discuss the epiTOC clocks have not, to our knowledge, been used to measure accelerated aging rate per se. We therefore have not included them in the discussion of the epigenetic aging clock.
- Line 292: Do you mean bisulfite? Line 306: We have edited bisulfate to bisulfite.

Reviewer 2 Report
The review by Larocca et al is a well written overview of three DNA based aging clocks highlighting many important examples and therapeutics strategies. Its concept is clear and relevant for a broad audience.
For general clarification of overall structure, it might however further improve by the additions of three headers for the three different described clocks. Each clock should be somewhat more extensively introduced with an explanation on how to measure the various clocks and be clearer on how these are affected during aging e.g., across the entire genome or at specific locations. Current review primarily focusses on how these clocks are related to lifespan of somatic cells while evidence of absence in germline as initially proposed in the introduction is largely lacking and should be more discussed.
An important point still lacking while comparing these three clocks, is what the correlation with human age is of each of the different clocks? Such values should be included and discussed on how accurately they could be of use for predicting biological age. Also, it could be worthwhile to indicate the rate of all three clocks during the various phases of aging within figure 2. Do these clocks change equally in males and females?
More specifically on the individual clocks:
Is the telomeric clock more a marker for the telomeric length at a given moment in time or is rather its decrease over time important?
This review contains at multiple sections some very nice historical aspects but unfortunately the first CpG methylation single gene clocks are lacking. This should still be incorporated prior to the description of the Horvath panel.
About the transposable element clock, does the amount of (R)TE differ per organism?
In addition to the three clocks presented and the proteomic and metabolomic clocks cited also clocks based on transcriptomics exists that most likely are influenced by and directly affect the genome. This should be added around line 475 jointly with the other clocks described.
With respect to therapeutic strategies the primary focus seems to be reprogramming factors. In the introduction and further throughout this review, it is inferred by the authors that the aging clocks are indications of reduced regenerative capacity and enhanced senescence. Is it known how these clocks are affected upon genetic or pharmaceutical interventions for the removal of senescent cells and if so how are these changed?
Lastly, there are some small grammar improvements needed such as abbreviations which are not properly introduced at the first appearances and Latin words such as in vivo that should be written in Italics.
Author Response
Reviewer 2: We appreciate the general comments of the reviewer. The evidence for absence of the clocks in the germline is discussed throughout the manuscript such as the resetting of the clocks by reprogramming by the egg cell in embryos, SCNT, and iPSC. We also discuss and reference the silencing of the TE clock in the germline by PIWI-piRNA pathway. Further evidence that the clocks are started at the initiation of differentiation is also discussed and referenced. With regard to location, the telomere clock is evident, the DNAm clock is discussed with regard to single loci and multiple loci and the focus on Polycomb repressive complex is discussed and referenced, the TE clock is dispersed throughout the genome and we reference how it can influence a diverse collection of genes. Prediction of biological age (acceleration and deceleration) is a theme that is discussed and referenced throughout the manuscript. In response to general comments on the structure, we added new headings as suggested to better describe the sections on the specific clocks. Below is our response to the specific suggestions:
- Is the telomeric clock more a marker for the telomeric length at a given moment in time or is rather its decrease over time important? We discuss the clock as a decrease in telomere length (accelerating or decelerating) in several parts of the manuscript Lines 128, 149-154, 159, 170, 172-175, specifically rate of shortening as it relates to lifespan, Lines 179-180.
- This review contains at multiple sections some very nice historical aspects but unfortunately the first CpG methylation single gene clocks are lacking. This should still be incorporated prior to the description of the Horvath panel. Lines 201-202: We have added the requested references to single or a few methylation loci clocks.
- About the transposable element clock, does the amount of (R)TE differ per organism? Although they are ubiquitous as we have stated the percentage does varies greatly between species of animals and plants.
- In addition to the three clocks presented and the proteomic and metabolomic clocks cited also clocks based on transcriptomics exists that most likely are influenced by and directly affect the genome. This should be added around line 475 jointly with the other clocks described. Lines 491-493: We have added a reference to recent transcriptomic clock.
- With respect to therapeutic strategies the primary focus seems to be reprogramming factors. In the introduction and further throughout this review, it is inferred by the authors that the aging clocks are indications of reduced regenerative capacity and enhanced senescence. Is it known how these clocks are affected upon genetic or pharmaceutical interventions for the removal of senescent cells and if so how are these changed? This is an interesting speculation, however an added discussion of senolytics is beyond the current scope of the manuscript.
- Lastly, there are some small grammar improvements needed such as abbreviations which are not properly introduced at the first appearances and Latin words such as in vivo that should be written in Italics. Abbreviations have been checked and Latin words italicized.

Reviewer 3 Report
In this manuscript, Larocca and colleagues review literature on somatic cellular clocks and their association with aging. In particular, they focus on the actions of telomere, DNA methylation, and transposable element. The authors also summarize the mechanisms contributing to the clock-free phenotype that observed in the germline cells. As the authors argue, chronological time can inform about underlying cellular pathology, as well as having potential as an intervention point for therapeutics. Each of these areas are subjects of intense study so this review is timely and relevant to the field. While the manuscript is well-written, I have a number of relatively minor concerns that should be addressed prior to publication.
Specific comments:
- Line 137, “Another type of immortal cell, the cancer cell, also expresses telomerase.” In the Ref#21 (Kim et al., Science, 1994), it was the activity of telomerase being examined in multiple cancer cell lines, not the levels of telomerase.
- Line 243, “Multiple DNAm clocks show accelerated aging in many disease conditions including cancer and they can successfully predict all-cause mortality and frailty (60, 61).” I feel very confused while reading this section, as the authors mention earlier that cancer cell is another type of immortal cell in Line 137. It would be helpful if the authors could modify this statement.
- Line 267, “Sir2, a member of a class of deacetylating enzymes called sirtuins”. It would be more appropriate to use: NAD+-dependent class III histone deacetylase.
- References are not correct in the following statements. Line 280, “As a consequence, age-related transcription of LINE elements and developmental genes Hoxa9 and Wnt occurs but can be prevented by over expression of SIRT1 and SIRT6 in aging mice (71).” Line 299 “Moreover, the flexibility of the clock is demonstrated by its deceleration in response to well-established aging interventions such as caloric restriction and mutation of growth hormone receptor (GhR) in mice (76).”
- Line 500 and 637, reprogramming factor should be c-myc.
Author Response
We are grateful for the careful reading of the manuscript and well thought out suggestions for minor revisions. We believe we have addressed the specific points and that the manuscript has been significantly improved upon further editing in response to the review. We have responded to the reviewer’s points as follows (Lines from reviewer’s comments refer to original, lines in our response refer to the revised manuscript):
- Line 137, “Another type of immortal cell, the cancer cell, also expresses telomerase.” In the Ref#21 (Kim et al., Science, 1994), it was the activity of telomerase being examined in multiple cancer cell lines, not the levels of telomerase. Lines 139: We have modified the sentence to correctly refer to “telomerase activity” in place of “expression of telomerase”.
- Line 243, “Multiple DNAm clocks show accelerated aging in many disease conditions including cancer and they can successfully predict all-cause mortality and frailty (60, 61).” I feel very confused while reading this section, as the authors mention earlier that cancer cell is another type of immortal cell in Line 137. It would be helpful if the authors could modify this statement. Lines 252-258: We have modified the statement with a brief discussion of this apparent paradox including 2 additional references - the Horvath clock in engineered hTERT expressing immortal cells and - a reference on senescence and epigenetic clocks.
- Line 267, “Sir2, a member of a class of deacetylating enzymes called sirtuins”. It would be more appropriate to use: NAD+-dependent class III histone deacetylase. Line 281-282: We changed the description of Sir2 to NAD+-dependent class III histone deacetylase.
- References are not correct in the following statements. Line 280, “As a consequence, age-related transcription of LINE elements and developmental genes Hoxa9 and Wnt occurs but can be prevented by over expression of SIRT1 and SIRT6 in aging mice (71).” Line 299 “Moreover, the flexibility of the clock is demonstrated by its deceleration in response to well-established aging interventions such as caloric restriction and mutation of growth hormone receptor (GhR) in mice (76).” Lines 292-296: The Oberdoerffer reference (now (81)) supports our statement. It is a mouse study showing SIRT1 has analogous RCM function as Sir2 in yeast, however, we have now included 2 additional references: (82) reports the SIRT6 KO accelerated aging in mice and (83) which provides evidence that SIRT6 has analogous RCM activity to Sir2 and that overexpression prevents dysregulated gene expression in response to DNA damage stress. Line 313-315: The original reference (now 88) supports our statement (see section entitled, “The rDNAm clock is responsive to interventions that modulate life-span” (supplemental figures S8 and S9))
- Line 500 and 637, reprogramming factor should be c-myc. Lines 517 and 655: We have changed the reprogramming factor accordingly.
